# Co-Infections and Superinfections between HIV-1 and Other Human Viruses at the Cellular Level

**DOI:** 10.3390/pathogens13050349

**Published:** 2024-04-24

**Authors:** Chiara Acchioni, Silvia Sandini, Marta Acchioni, Marco Sgarbanti

**Affiliations:** Department of Infectious Diseases, Istituto Superiore di Sanità, Viale Regina Elena 299, 00161 Rome, Italy; chiara.acchioni@iss.it (C.A.); silvia.sandini@iss.it (S.S.); marta.acchioni@iss.it (M.A.)

**Keywords:** co-infection, superinfection, HIV, HTLV, HHVs, flaviviruses, EBOV, CD4^+^ T cells, macrophages, DCs

## Abstract

Co-infection or superinfection of the host by two or more virus species is a common event, potentially leading to viral interference, viral synergy, or neutral interaction. The simultaneous presence of two or more viruses, even distantly related, within the same cell depends upon viral tropism, i.e., the entry of viruses via receptors present on the same cell type. Subsequently, productive infection depends on the ability of these viruses to replicate efficiently in the same cellular environment. HIV-1 initially targets CCR5-expressing tissue memory CD4^+^ T cells, and in the absence of early cART initiation, a co-receptor switch may occur, leading to the infection of naïve and memory CXCR4-expressing CD4^+^ T cells. HIV-1 infection of macrophages at the G_1_ stage of their cell cycle also occurs in vivo, broadening the possible occurrence of co-infections between HIV-1 and other viruses at the cellular level. Moreover, HIV-1-infected DCs can transfer the virus to CD4^+^ T cells via *trans*-infection. This review focuses on the description of reported co-infections within the same cell between HIV-1 and other human pathogenic, non-pathogenic, or low-pathogenic viruses, including HIV-2, HTLV, HSV, HHV-6/-7, GBV-C, Dengue, and Ebola viruses, also discussing the possible reciprocal interactions in terms of virus replication and virus pseudotyping.

## 1. Introduction

### 1.1. General Features of Viral Co-Infection/Superinfections

Co-infection of an individual with two or more viruses belonging to different families, genera, or species can occur [1]. Co-infections can involve viruses causing either chronic or acute infections or a combination of both. The simultaneous presence of different viruses is quite common and can result in different outcomes. An intriguing question is what happens when different viruses (usually two) infect the same cell, either at the same time (co-infection) or one after the other (sequential infection or superinfection), even if these definitions better apply to the same individual rather than the same cell. Interference is one possibility where both viruses compete for the host cell resources. During co-infection or sequential infection, direct inhibition can occur, exerted by one virus over the replication of the other by a specific molecular mechanism, as in the case of the suppression of hepatitis B virus (HBV) enhancers 1 and 2 by the hepatitis C virus (HCV) core protein [2,3]. During a dual infection, a viral product belonging to a specific virus can even promote the replication of a different co-infecting heterologous virus, as in the case of Epstein–Barr virus (EBV)-encoded nuclear antigen 1 (EBNA1), capable of enhancing HCV replication [4]. Interference also involves a protective interferon (IFN)-mediated innate immune response, even when, during a sequential infection, the first virus is inactivated [5]. The latter case is evident for respiratory viruses, where dual co-infection or sequential infection of influenza A virus (IAV) H1N1 or respiratory syncytial virus A (RSV-A) with rhinovirus A16 (RV-A16) determines RV-A16 inhibition of replication due to the high sensitivity of the latter virus to IFN-I (α and β) and IFN-III (λ) [6]. During co-infection/superinfection, a virus fully competent for replication can assist the replication of a defective virus. This is the case for Hepatitis D virus (HDV), which depends upon cellular co-infection with a helper virus like HBV. HDV can transcribe its RNA and produce the capsid ribonucleoprotein, but cannot associate with the cell membrane, and infectious particles cannot be released from infected cells unless the HBV membrane glycoprotein HbsAg is present [7].

### 1.2. HIV Viruses

HIV viruses belong to the *lentivirus* genus of the *Retroviridae* and cause persistent infections characterized by the integration of their proviral genomic DNA (generated upon reverse transcription of genomic RNA by the viral reverse transcriptase) into the host genome, mostly in the proximity of introns of genes under active transcription [8]. The Human Immunodeficiency Virus (HIV)-1 pandemic made its global appearance more than 40 years ago, and has since then caused about 40.4 million deaths. An estimated 33.1 to 45.7 million people were living with HIV in 2022 (https://www.unaids.org/en/resources/fact-sheet, accessed on 26 February 2024). HIV-2, a related yet immunologically distinct relative of HIV-1, was discovered in 1986 in West African patients and associated with a milder disease compared to HIV-1 [9]. Through the binding of the viral Env glycoprotein gp120 to the CD4 receptor, followed by interaction with either CCR5 or CXCR4 cellular co-receptors and subsequent membrane fusion mediated by viral gp41, HIV-1 is able to productively infect CD4^+^ T lymphocytes, resulting in a latent infection in distinct CD4^+^ T cell differentiation stages [10]. Replication of the HIV-1 virus, as well as reactivation of viral transcription from latency, requires the presence and activity of the HIV-1 *trans*-activator of transcription (Tat) protein and regulator of virion expression (Rev). The Tat protein binds both the cyclin (Cyc)T1 subunit of the P-TEFb elongation complex and the transactivation-responsive element (TAR) at the 5′ end of HIV transcripts, dramatically increasing the rate of transcription elongation by RNA Polymerase (Pol) II [11,12]. The Rev protein, on the other hand, is a sequence-specific RNA-binding protein responsible for the nuclear export of intron-containing HIV-1 RNAs [13]. Several accessory viral proteins are also expressed during HIV-1 replication, such as virion infectivity factor (Vif), viral protein U (Vpu), viral protein R (Vpr), and negative regulatory factor (Nef), involved in viral evasion from innate and adaptive immune responses, and efficient replication in non-dividing cells [14,15]. The HIV-1 proviral genomic structure presents the typical retrovirus organization with the *gag* gene (coding for capsid, nucleocapsid, and matrix proteins), *pol* gene (coding for reverse transcriptase, protease, and integrase), and *env* gene (coding for envelope glycoproteins gp120 and gp41) within the long terminal repeats (LTR) at the 5′ and 3′ ends. The 5′ LTR sequence represents the virus promoter that controls HIV-1 transcription, initiated primarily by activated cellular transcription factors and then dramatically enhanced by Tat [16].

### 1.3. Potential HIV-1 Co-Infecting/Superinfecting Viruses at the Cellular Level

Co-infection or superinfection of an HIV-1-infected CD4^+^ T cell may occur when two viruses with the same cellular tropism enter the same cell together or one after the other, respectively. Therefore, either HIV-1 viral particles belonging to different HIV-1 strains, or HIV-1 and HIV-2 viruses, may be present in the same cell together [17,18].

Another possible scenario is the co-infection or sequential infection of HIV-1 and human T-cell lymphotropic virus (HTLV) types 1 and 2.

Surprisingly, human herpes simplex viruses (HSV)-1 and -2 can also end up in the same cell with HIV-1, mostly because of HSV’s broad cellular tropism, resulting in the modulation of HIV-1 replication and the production of pseudotyped HIV-1 virus particles [19,20]. Other human herpesviruses (HHVs), like the lymphotropic HHV-6 and HHV-7, can also infect CD4^+^ T cells in vivo, increasing HIV-1 pathogenicity in HHV-6 or HHV-7 and HIV-1 co-infected cells [21]. Often, HIV-1 patients are co-infected with another HHV, the human herpesvirus-8 (HHV-8), previously known as Kaposi’s sarcoma herpesvirus (KSHV). Elongated, spindle-like, HHV-8-infected endothelial cells are the most important component of Kaposi’s sarcoma (KS) lesions. Importantly, AIDS-associated KS is typical of the terminal stages of HIV infection. HHV-8 is also capable of latently infecting B cells and is responsible for B lymphoproliferative diseases, including primary effusion lymphoma (PEL) and the plasmablastic variant of multicentric Castleman disease (MCD), both found with a higher frequency in HIV-1 patients [22]. Even if HIV-1/HHV-8 co-infection in patients is clinically relevant, it is unlikely that co-infection also occurs within the same cell due to the different tropisms of the two viruses.

Unexpectedly, GB virus C (GBV-C), a human non-pathogenic lymphotropic flavivirus, co-infects HIV-1 target cells, resulting in lower HIV RNA levels, slower progression to AIDS, improved survival, and higher CD4^+^ T cell counts in co-infected patients [23].

An important feature of HIV-1, favoring co-infection/superinfection with related or unrelated viruses, is its capacity to establish latent infections, especially in long-lived CD4^+^ resting memory T cells, thus providing a cellular reservoir infectable by other lymphotropic viruses.

Moreover, HIV-1 can also productively infect antigen presenting cells (APCs), such as macrophages, while HIV-1-infected dendritic cells (DCs) can transfer the virus to CD4^+^ T cells [24,25]. Therefore, HIV-1 co-infecting/superinfecting pathogenic viruses are not limited to those endowed with CD4^+^ T cell tropism. This review analyzes possible outcomes of co-infections/superinfections, reported in the literature, between HIV-1 and other human pathogenic, non-pathogenic, or low-pathogenic viruses occurring within the same cell and the possible reciprocal interactions affecting the replication of either HIV-1 and the co-infecting/superinfecting viruses, as well as virus pseudotyping, a phenomenon that can broaden HIV-1 cellular and tissue tropism. Other potential co-infections involving viruses for which established vaccinations have been available for a long time, such as influenza viruses (IAVs), measles virus (MeV), and yellow fever virus (YFV), were not included, or only briefly described, in this review.

## 2. Co-Infections or Superinfections of HIV-1 with Human Pathogenic Viruses Targeting CD4^+^ T cells

### 2.1. Co-Infection/Superinfection between Different HIV-1 Strains

In the case of HIV-1, co-infections of individuals are defined as double or multiple infections with phylogenetically different HIV-1 strains, happening almost simultaneously and before seroconversion, while superinfections are defined as events happening after seroconversion [26]. At the cellular level, a better definition of superinfection is the infection of a cell harboring an already integrated proviral DNA (either in a transcriptionally active or silent state) belonging to the first virus that infected that cell. The outcome in this case, as well as in the case of a simultaneous infection (co-infection at the cellular level), is the integration of 2 or more proviruses from different HIV-1 strains [27].

One of the main features of HIV-1 is its extremely elevated genetic diversity that led to the use of the term quasispecies to describe the heterogeneity of the diverse virus populations within a single infected individual [28].

HIV-1 has been classified into four groups: M (main), O (outlier), N (non-M, non-O), and the more recent P (named as such because P is the first letter coming after O in the alphabet) [29,30]. M, the most abundant, is composed of nine subtypes (A, B, C, D, F, G, H, J, and K) and about 157 circulating recombinant forms (CRFs), according to the HIV Sequence Database (http://www.hiv.lanl.gov/content/sequence/HIV/CRFs/CRFs.html, accessed on 27 February 2024). Several unique recombinant forms (URFs) were also detected in single individuals only and, by definition, in less than three epidemiologically unlinked patients [29,31].

CRFs and URFs demonstrate the possibility of co-infections or superinfections within the same cell. It is feasible that co-infections or superinfections involving viruses in active replication generate new virus particles that may be genetic chimeras containing parts of both parental viruses. In this respect, the first strong evidence of lentivirus recombination in nature was detected by studying the simian immunodeficiency viruses in West African green monkeys (SIV_AGM_). These viruses were characterized by a mosaic genome structure with features belonging to SIV isolated in sooty mangabeys (SIV_SM_), thus suggesting recombination upon cross-species transmission [32]. Indeed, the diploid nature of HIV-1 virions, as well as reverse transcriptase (RT) switching between the two different RNA strands, both contribute to generating genetic hybrids (Figure 1) that may also be relevant for HIV-1 genome repair, thus rescuing defective viruses [18,33].

Pseudotyping between resident HIV-1 (HIVHXB2), persistently infecting the H9 cell line (resulting in H9HTLVIIIB cells), and superinfecting HIV-1 (HIVMN) was also demonstrated in vitro, resulting in virus particles harboring the HIVHXB2 genome and the Env protein of the HIVMN virus, from day 1 to day 6 post-superinfection, resulting in enhanced viral replication [34]. HIV-1 pseudotype production due to superinfection, may mediate in vivo immune escape and viral persistence. On the other side, in the case of HIV-1 superinfection of HIV-1 latently infected cells, for these viruses to produce recombinants, it is necessary that the latent virus reactivates its transcription and replication. In this respect, it has been demonstrated, using latently infected Jurkat T cell lines as well as primary CD4^+^ T cells, that HIV-1 superinfection results in the reactivation of latent HIV-1, depending on the expression of the Tat protein from the superinfecting virus [35].

The CD4^+^ T cell reservoir of HIV-1 latently infected cells is estimated to be between 1 and 1000 cells per million, most likely 10–100 [36]; therefore, infection with a second HIV-1 virus, possibly due to therapy interruption or non-adherence to therapy, most probably will involve uninfected CD4^+^ T cells. Moreover, by using single-genome analysis from HIV-1-infected patients, it has been estimated that only 2.4% of analyzed genomes were intact, while 97.6% had fatal mutations/deletions potentially affecting proviruses’ transcription upon latency reversal [37]. However, the situation may be very different in secondary lymphoid organs, like in the case of the spleen from infected patients, where an average of three or four proviruses per infected cell were detected by fluorescence in situ hybridization, and evidence was found for large numbers of recombinants coupled with genetic variation [38]. By examining the rate of HIV-1 recombination using two vectors expressing different strains of HIV-1 and a single replication cycle, it was possible to measure recombination events per genome per replication cycle, and those were estimated as two to three [18], about tenfold higher than point substitutions of about 0.25 per genome per replication cycle [39], indicating a potentially more relevant impact of co-infection/superinfection-mediated recombination in HIV-1 genetic variability, immune escape, pathogenicity, and antiretroviral therapy failure due to pan-resistant viruses that can be generated starting from multiple drug-resistant (MDR) strains. [40].

One of the obstacles to this scenario is represented by superinfection resistance (SIR), where superinfection of the same cell is inhibited early after the first infection, usually between 4 and 24 h [41]. The main mechanism of SIR is the downmodulation of the CD4 receptor during a productive HIV-1 infection of CD4^+^ T cells [41]. Transmembrane CD4 cellular receptor for HIV is present on the surface of CD4^+^ T cells, macrophages, and DCs. The binding of the HIV-1 envelope glycoprotein gp120, protruding from the envelope of HIV-1 viral particles, with CD4 produces conformational modifications of gp120, leading to its interaction with the CCR5 or CXCR4 cellular coreceptor, with the subsequent membrane fusion, mediated by HIV-1 glycoprotein gp41, and HIV-1 entry into target cells [27]. Different viral proteins, such as Nef, Vpu, and gp160 precursor of gp120 and gp41, have been shown to affect CD4 presence at the cell membrane. While Nef causes CD4 endocytosis, both Vpu and gp160 impede CD4 accumulation at the cell membrane through different mechanisms [42]. The physiological role of CD4 downmodulation subsequent to HIV-1 infection is to facilitate the release of virus particles from the cell membrane [43]. Nevertheless, HIV-1 viruses using the CCR5 co-receptor, characteristic of the early stages of infection, do not downmodulate CD4, differently from viruses entering through CXCR4 or CXCR4/CCR5, which are common during AIDS [44]. Accordingly, different nef alleles from asymptomatic patients were far less capable of downmodulating CD4 compared to those from AIDS patients when both were substituted for the wt nef gene of the HIV-1 X4-tropic NL4.3 virus [43]. Less clear is the contribution of co-receptor downmodulation to SIR [41]. CD8^+^ T cell non-cytotoxic antiviral responses (CNAR) and the CD8^+^ T cell antiviral factor (CAF) inhibit HIV replication without target cell killing but do not prevent superinfection, reverse transcription, and provirus integration; rather, they inhibit Tat-mediated viral transcription from the HIV-1 long terminal repeats (LTR) promoter [45].

### 2.2. Co-Infection/Superinfection between HIV-1 and HIV-2

HIV-2 represents a lentivirus related to HIV-1, sharing a similar genomic structure and derived from simian immunodeficiency virus (SIV) isolated from sooty mangabey (SIV_SM_), differently from HIV-1, which is derived from SIV infecting chimpanzees (SIV_cpz_) [46]. Both viruses co-circulate in Western Africa, and HIV-2 causes a milder disease compared to HIV-1, characterized by a lower plasma viral load and infectivity [47] and a slower decline of CD4^+^ T cells and progression to AIDS [48,49]. Among the mechanisms proposed for HIV-2′s lower pathogenicity, there is the capacity of naïve dendritic cells to become activated upon infection with HIV-2, eliciting IFN-I responses [50]. Indeed, HIV-2 is able to efficiently replicate in DCs, differently from HIV-1, due to the activity of its Vpx protein on the cellular restriction factor alpha-motif (SAM) and histidine-aspartate (HD) domain-containing protein 1 (SAMHD1), inhibiting the SAMHD1-mediated hydrolysis of deoxynucleoside triphosphates (dNTPs) necessary for reverse transcription [51]. HIV-2 Vpx is present within newly released virions [52,53], and determines the degradation of SAMHD1 upon entry [51,54]. Another mechanism by which HIV-2 infection of DCs stimulates IFN-I production is the recognition of HIV-2 retrotranscribed cDNA by the cyclic GMP-AMP synthase (cGAS) host sensor due to the inefficient shielding capacity of the HIV-2 capsid [50]. Subsequently, HIV-2-infected DCs are able to stimulate a CD4^+^ T cell response against the virus [55]. More importantly, HIV-2 seems to reduce HIV-1-mediated pathogenesis in patients infected with both viruses due to cross-protective antibodies and cytotoxic CD8^+^ T cell-mediated responses elicited by the HIV-2 infection [56,57]. HIV-1 and HIV-2 can also co-infect the same cell when dual infections are performed in cell culture, and in this case, HIV-2 is able to inhibit HIV-1 replication in a selective, dose-dependent, and nonreciprocal fashion [58]. Indeed, HIV-2 is not able to inhibit the expression of the Rauscher murine leukemia virus (R-MuLV) provirus and R-MuLV replication, while a dose-response inhibition of HIV-2 (SXb-1) provirus over HIV-1 (IIIB) provirus was obtained upon co-transfection of CD4^+^ T lymphocytic CEM cells, with more than 90% inhibition at a 0.5 molar ratio, and finally, HIV-1 was not able to inhibit HIV-2 gene expression [58]. Different molecular mechanisms are involved, including the interaction between HIV-2 Rev and the Rev responsive elements RNA from HIV-1 (RRE1), resulting in poor or aberrant HIV-2 Rev dimerization during the binding to RRE1 [59]. Another suggested mechanism points to the HIV-2 Vpx protein as an inhibitor of HIV-1 replication. When HIV-1 chimeric viruses were generated, expressing a functional or non-functional Vpx protein instead of the HIV-1 Vpr protein, productive infection of the CD4^+^ T cell line Sup-T1 was obtained only in the absence of Vpx expression [60]. Finally, HIV-2 inhibition of HIV-1 was mapped within the HIV-1 LTR promoter [58,61]. In particular, the HIV-2 TAR element, the target of HIV-1 and HIV-2 Tat proteins, was implicated in HIV-2-mediated inhibition of HIV-1 replication through a decoy effect, demonstrated following stable nucleolar expression in CEM cells [62]. The stretch of the HIV-2 TAR sequence, responsible for HIV-1 inhibition, was mapped within the second stem loop, acting by causing RNA polymerase II complexes to stall, thus inhibiting HIV-1 transcription elongation [63]. Despite HIV-2-mediated inhibition of HIV-1 co-infection or superinfection, HIV-1 superinfection of HIV-2 CD4^+^ T cells was obtained in vitro, generating pseudotyped virions characterized by expanded cellular range [17].

### 2.3. Co-Infection/Superinfection between HIV-1 and HTLV

HTLV-1/2 belong to the *deltaretrovirus* genus of the *Retroviridae* family, and HTLV-1, apart from being the first human retrovirus discovered [64,65], is also the etiological agent of a fatal and aggressive disease known as human adult T cell leukemia/lymphoma (ATL) [66] and a chronic neurologic inflammatory disease, myelopathy/tropical spastic paraparesis (HAM/TSP) [67]. HTLV-1/2 are capable of infecting and immortalizing CD4^+^ and CD8^+^ T cells in vitro [68], and these cells are HTLV targets in infected individuals [69,70].

There are no specific biological markers or clinical symptoms that can forecast the development of HTLV-1-related diseases; nevertheless, HTLV-1 proviral load has been proposed. Worldwide, it is estimated that there are 5 to 10 million people living with HTLV-1, even though it is likely an underestimation (https://www.who.int/news-room/fact-sheets/detail/human-t-lymphotropic-virus-type-1, accessed on 11 March 2024). Considering HTLV-1 distribution worldwide, transmission routes (sexual contact, breastfeeding, blood transfusion, and intravenous drug use), and HTLV-1 target cells, including CD4^+^ T cells, it is not surprising that patients infected with both viruses were found, with co-infection promoting a higher incidence of HAM/TSP in the co-infected patients compared to the mono-infected ones (15.7% and 9.7%, respectively) [71,72], but with increased HIV-1 pathogenicity and discrepancy between CD4^+^ T cell count and the immunocompetent status of the infected patients in some studies [73,74], and an increased trend of survival in another study, though not statistically significant [75]. Differently from HTLV-1, HTLV-2 co-infection with HIV-1 was associated with slower HIV-1 progression [75]. HTLV-1 has a broader spectrum of potential target cells compared to HIV-1, using glucose transporter (GLUT)-1, neuropilin, and heparan sulfate proteoglycan, as potential receptors, even though this is still a matter of debate [76,77,78,79]. Therefore, HTLV-1 is also able to infect CD8^+^ T cells, monocytes, B cells, and epithelial cells in vivo [80,81]. Soluble factors, like the CC chemokines RANTES (regulated on activation, normal T cell expressed and secreted), macrophage inflammatory protein (MIP)-1α, and MIP-1β, secreted by HTLV-1-infected cells, were shown to inhibit infection of HIV-1-permissive cells by M-tropic viruses while promoting T-tropic virus infection, differently from soluble Tax protein, which was shown to promote the infectivity of both types of viruses, particularly in synergy with TCR engagement [82]. In this respect, CC chemokines are CCR5 ligands, and uninfected individuals continuously exposed to HIV-1 have higher levels of these chemokines [83]. Such in vitro experiments suggested that in HTLV-1/HIV-1 co-infected patients, the transition from an M-tropic to a T-tropic HIV-1 virus is facilitated, thus potentially driving a faster onset of AIDS in the absence of therapy [82]. Due to the fact that both HTLV-1 and HIV-1 can target CD4^+^ T cells for productive infection, co-infection or superinfection may also occur at the cellular level in co-infected patients. Indeed, in vitro co-infection/superinfection of CD4^+^ T cells with HIV-1 and HTLV-1 can be obtained, resulting in pseudotyping of HIV-1 virions with HTLV-1 Env [84]. At the same time, reciprocal pseudotyping was also demonstrated, with HTLV-1 virus bearing HIV-1 Env being able to replicate and compete with wt HIV-1, therefore mediating superinfection exclusion [84]. The outcome of HTLV-1 and HIV-1 pseudotyping, particularly the production of HIV-1 virions with HTLV-1 Env, can broaden the spectrum of HIV-1 target cells; in this case, only a single round of infection can be completed, unless there is a concomitant superinfection with wt HTLV-1, able to produce new pseudotyped HIV-1 virions. This is the case of co-infection of primary female vaginal and cervical epithelial cells by HIV-1 pseudotyped with HTLV-1 Env and HTLV-1, both coming from dually infected cells [85]; as a result, there are new cycles of infection of these cells by pseudotyped HIV-1 [81]. In an in vivo setting, sexually transmitted pseudotyped HIV-1 and HTLV-1, both coming from dually infected patients, have greater chances of being further transmitted to intraepithelial CD4^+^ T cells, macrophages, or DCs [81].

### 2.4. Co-Infection/Superinfection between HIV-1 and HSV

Human herpes simplex viruses (HSV-1 and -2) possess a broad spectrum of target cells, entering their host via mucosal epithelia, skin, or cornea and then moving to the nervous system, where they are able to establish latent infections of neurons [85]. HIV-1 promotes acquisition and reactivation of HSV, and HSV favors HIV-1 infection and causes severe clinical manifestations in immunocompromised hosts, particularly in AIDS patients [86,87,88,89]. More importantly, HSV infection can worsen HIV-1 pathogenesis by promoting HIV-1 replication [88]. It has been shown that HSV can infect activated primary human peripheral blood lymphocytes (PBL) and CD4^+^ T cell lines [90]. Using HSV-1 deletion mutants of immediate-early proteins, ICPO and ICP27 were identified as essential for the upregulation of HIV replication [19,91], with an initial increase in subgenomic (2.0 and 4.3 kb) mRNA accumulation, followed by unspliced genomic (9.2 kb) mRNA [19]. In particular, ICPO cooperates with the HSV-1-activated NF-κB cellular transcription factor in reactivating HIV-1 transcription from ACH-2 CD4^+^ T cell latency model [92]. In addition, the HSV-1 US11 late gene product, present within virus particles, has been shown to cooperate with suboptimal concentrations of Rev onto Rev responsive element (RRE)-containing mRNAs to promote HIV-1 Env protein translation and behave as a Rev auxiliary protein when Rev accumulates below a threshold necessary for HIV-1 replication [93,94]. Another important outcome of HSV and HIV-1 co-infection at the cellular level is the pseudotyping of HIV-1 virus particles with HSV envelope [90], resulting in an extended range of HIV-1-infectable cells. Indeed, CD4-negative epidermal keratinocytes from co-infected AIDS patients without genital HSV-mediated lesions were observed to be infected by HIV-1, possibly pseudotyped with HSV Env [20]. Therefore, a reciprocal enhancement of viral replication can be observed between HIV-1 and HSV-1 upon co-infection of the same cells [20].

## 3. Co-Infections or Superinfections of HIV-1 with Low-Pathogenic, or Non-Pathogenic, Human Viruses also Targeting CD4^+^ T Cells

### 3.1. HIV-1 and HHV-6/HHV-7

Human herpes viruses 6 (HHV-6) and 7 (HHV-7) are lymphotropic β-herpesviruses responsible for the acute illness *Roseola infantum* in early childhood [21]. These low-morbidity viruses are persistent and kept under control in immunocompetent individuals, but they can behave as opportunistic agents in immunocompromised hosts. In this respect, clinical and in vitro studies have indicated that HHV-6 (particularly HHV-6A) is a cofactor in HIV-1 disease progression [21,95]. Both HHV-6 and HHV-7 are able to infect CD4^+^ T cells in vitro and in vivo, with HHV-6 preferentially infecting activated cells [95]. HHV-6 exerts a direct cytopathic effect on CD4^+^ T cells [96] and is able to determine transactivation of the HIV-1 LTR, also through a synergistic interaction between HHV-6 proteins and HIV-1 Tat [97,98,99]. Moreover, HHV-6 infection determines the expression of CD4 on CD8 T cells and NK cells [100,101], stimulating the CD4 promoter [102], thus expanding HIV-1 tropism. HHV-6 infection also promotes the downregulation of CD3 and the complement regulatory protein CD46, the HHV-6 receptor, resulting in T-cell function dysregulation [103,104]. It is assumed that, upon HHV-6 reactivation, even more unbalanced immune responses are obtained in HIV-1 co-infected patients. In this respect, HHV-6 also non-productively infects macrophages and DCs, resulting in the downregulation of the Th1-promoting cytokine IL-12 [95]. Moreover, HHV-6 infection induces the production of RANTES, which inhibits the replication of R5 HIV-1, possibly promoting the switch to X4 HIV-1 viruses [105]. HHV-7, on the contrary, causes the downregulation of CD4, its receptor, without augmenting RANTES production. In HHV-7/HIV-1-coinfected human lymphoid tissue, CD4 downregulation suppresses the replication of HIV-1 R5 viruses but not X4 viruses, a selective effect explained by HHV-7 being, in turn, suppressed by X4 viruses, a phenomenon that prevents CD4 downregulation [104,106]. In conclusion, a major in vivo outcome of both HHV-6 and HHV-7 is the change from R5 to X4 viruses, resulting in an accelerated progression of HIV-1-mediated disease in the absence of therapy.

### 3.2. HIV-1 and GBV-C

GB virus C (GBV-C), with six different genotypes is a human non-pathogenic lymphotropic virus member of the *Flaviviridae* family, assigned to the proposed *Pegivirus genus* [107]. GBV-C was revealed in peripheral blood mononuclear cells (PBMCs), spleen, and bone marrow and shown to replicate in PBMCs and primary B and T lymphocytes in vitro [23]. GBV-C infection occurs by sexual contacts, parenteral blood or blood product exposure, and mother-to-child vertical transmission [23]. GBV-C is distributed worldwide, and among people living with HIV-1, HCV, or HBV infection, its prevalence can reach up to 50% [23]. GBV-C is usually cleared from infected individuals, but a persistent infection can also occur, lasting for decades [108,109]. Clinical studies provided evidence that HIV-1 patients co-infected with GBV-C showed lower HIV-1 RNA levels, slower progression to AIDS, improved survival, and higher CD4^+^ T cell counts compared to patients infected with HIV-1 only [110,111,112,113,114,115]. Moreover, an inverse correlation was found between GBV-C and HIV viral load, with increased median GBV-C RNA levels after HAART initiation, followed by a decrease upon HAART interruption and subsequent restoration of HIV-1 replication [115], thus suggesting a reciprocal inhibition of replication between the two viruses. These beneficial effects were pronounced in HIV-1 patients with advanced disease and low CD4^+^ T cell count (<200/μL) [113,116] and in patients co-infected with GBV-C genotype 2, subtype b (2b) [117]. Different mechanisms were proposed to account for GBV-C’s action over HIV-1 pathogenesis. Circulating CD80^+^ plasmacytoid dendritic cells (pDCs), correlating directly with higher CD4^+^ T cell counts and inversely with viral load in HIV-1 patients [118], are increased in number and activation state in GBV-C/HIV-1 co-infected patients [119]. Indeed, high HIV-1 viral load was associated with increased pDC cell death by apoptosis and necrosis [120]. Moreover, in vitro GBV-C/HIV-1 co-infection of PBMCs results in increased expression and release of SDF-1, RANTES, MIP-1α, and MIP-1β chemokines, able to bind CXCR4 (SDF-1) and CCR5 (RANTES, MIP-1α, and MIP-1β) HIV-1 co-receptors, thus causing both co-receptors’ downmodulation at the cell surface [121]. More importantly, antibodies against these chemokines were able to eliminate the inhibition of HIV-1 replication by GBV-C in vitro [121]. GBV-C envelope protein 2 (E2) was shown to inhibit HIV-1 entry by binding to and interfering with HIV-1 gp41 fusion protein through the modification of its conformation [122,123]. A reduced chronic deleterious activation of CD4^+^ and CD8^+^ T cells, independent of HIV-1 viral load or CD4/CD8 ratio, was also observed in GBV-C/HIV-1 co-infected patients [124]. Finally, in the absence of HAART treatment, a lower expression of Fas (CD95/Apo-1) and reduced Fas-mediated apoptosis was detected on the cell surface of PBMCs derived from co-infected patients compared to patients infected with HIV-1 only [125].

## 4. Co-Infections or Superinfections of HIV-1 with Human Pathogenic Viruses Targeting Macrophages and DCs

### 4.1. HIV-1 and Dengue

Infections with the four antigenically distinct serotypes (1, 2, 3, and 4) of *Dengue virus* (DENV), a member of the *Flaviviridae* family, cause the most important arboviral disease in humans [126]. It is estimated that there are 3.9 billion people at risk of contracting DENV worldwide (https://www.who.int/news-room/fact-sheets/detail/dengue-and-severe-dengue, accessed on 15 March 2024), and in January 2024, over half a million dengue cases and over 100 dengue-related deaths were already reported globally (https://www.ecdc.europa.eu/en/dengue-monthly, accessed on 15 March 2024). Therefore, it is feasible to assume that it is highly probable to have co-infections/superinfections between HIV-1 and DENV in areas of the world where DENV is endemic. Indeed, cases of co-infections were reported [127,128,129,130,131]. From a dengue disease perspective, co-infection with HIV-1 mostly resulted in a mild disease [127,128,131], or when severe dengue was present, no deadly cases of Dengue hemorrhagic fever (DHF) or Dengue shock syndrome (DSS) were reported [130,132]. Interestingly, no increased HIV-1 viral load, or worsening of HIV-1-related clinical symptoms were observed in co-infected patients [127,128,131]. On the contrary, a report actually described a transient inhibition of HIV-1 replication both in vivo and in vitro, with a decrease in HIV-1 viral load during the acute phase of DENV infection and an inhibitory action of acute-phase serum from different HIV-1-uninfected dengue patients on PBMCs from healthy donors infected with HIV-1 [127], similarly to what was reported with measles virus/HIV-1 co-infection [133]. Both HIV-1 and DENV can productively infect macrophages [24,134], and tissue-resident macrophages are considered one of the HIV-1 cellular reservoirs in infected patients under antiretroviral therapy [135]. Therefore, it is possible to have both viruses within the same cell in co-infected patients. In this respect, the role of NS5 proteins belonging to different flaviviruses, including DENV NS5, was proposed to inhibit HIV-1 replication when co-expressed in HIV-1-infected cells through CD4 receptor downmodulation [136,137], while yellow fever virus (YFV) infection of macrophages was shown to inhibit HIV-1 replication in infected cells, also correlating with CD4 downmodulation [137].

### 4.2. HIV-1 and Ebola Virus

Ebola viruses (EBOV) belonging to the *Filoviridae* family are emerging viruses determining outbreaks of human hemorrhagic fever [138]. The largest Zaire Ebola outbreak in West Africa, recorded to date from 2014 to 2016, increased the chances of the occurrence of HIV-1/EBOV co-infections [139]. In a case report, a patient dually infected with HIV-1 and HIV-2, under antiretroviral treatment and a normal CD4^+^ T cell count, showed persistently detectable and infectious EBOV in semen 565 days after recovery from Ebola virus disease (EVD), suggesting a possible involvement of HIV-mediated chronic inflammation and immune dysregulation/senescence in EBOV persistence [140]. EBOV is capable of productively infecting DCs and tissue macrophages early upon infection [141], while HIV-1 can only productively infect macrophages, particularly tissue macrophages [135], and DCs do not replicate HIV-1 but can *trans*-infect CD4^+^ T cells [24,25]. Therefore, in a diverse fashion, macrophages and DCs may harbor both viruses in co-infected patients. Interestingly, HIV-1 can bind to the sialic acid-binding Ig-like lectin 1 (Siglec-1/CD169) DC cell receptor [142,143], similarly to EBOV, which uses the Siglec-1 receptor to enter the DC cytoplasm [144]; monoclonal antibodies (mAbs) elicited against Siglec-1 cross-protect against EBOV and HIV-1 attachment to activated DCs, EBOV cytoplasmic entry, and activated DC-mediated HIV-1 *trans*-infection of CD4+ T cells [144].

## 5. Discussion

Viral co-infections/superinfections of a host may actually be the rule instead of the exception. Evidence from dated as well as recent papers points to co-infections and superinfections of the same tissue, cell types, or even the same single cell as something happening in vivo and also reproducible *in vitro*. In this respect, the possibility of having HIV-1-infected cells co-infected or superinfected by closely or distantly related viruses depends upon the possibility of productively infecting the same cell targets of HIV-1 by these viruses. While this is obvious for different HIV-1 groups or subtypes (also determining the appearance of recombinant viruses), HIV-2, and HTLV-1 and -2, it is not immediately apparent for more distant viruses, like HHVs, GBV-C (Figure 2), and DENV or EBOV (Figure 3).

While, for example, HSV’s broad spectrum of potential target cells can provide a window of opportunity within CD4^+^ T cells, the ability of HIV-1 to productively infect macrophages and to trans-infect CD4^+^ T cells through DC-mediated non-productive infection, offers the ideal environment for DENV and EBOV concomitant presence with HIV-1. Co-infection, and superinfection between HIV-1 and other viruses at the cellular level may have beneficial or detrimental effects for both viruses, the cell they infect, and the infected host as a whole. Another important aspect to be considered is that the presence of different viruses together with HIV-1 within the same cell may lead to HIV-1 pseudotyping, thus further increasing the spectrum of HIV-1-infectable cells.

Co-infection or superinfection of the same cells with genetically distinct HIV-1 viruses, giving birth to possible recombinants endowed with increased resistance to antiretroviral drugs, becomes relevant for the management of co-infected patients [30]. One of the main problems derives from the fact that the design of antiretroviral drugs, the research concerning drug resistance mutations, and the development of assays for drug resistance detection were largely based on HIV-1 subtype B because of its initial prevalence in highly developed countries where such drugs were first introduced. However, subtype B viruses only account for 10% of all circulating subtypes, and non-B subtypes have been increasing in developed countries, including complex inter-subtype recombinants [145]. Moreover, the development of specific mutations varies among different subtypes due to the intrinsic properties of the virus and not only by different pressure from antiretroviral drugs [146]. In this respect, resistance to nucleoside reverse-transcriptase inhibitors (NRTIs), non-nucleoside reverse transcriptase inhibitors (NNRTIs), and protease inhibitors (PIs) can differ between subtypes [147]. Among non-B subtypes, subtypes C and D were found to be the most aggressive, while A was the least, and in the middle were subtypes G, AE, and AG in descending order of pathogenicity [30]. However, the recombinant HIV-1 CRF01_AE strain possesses an increased risk of faster disease progression, characterized by a faster rate of CD4^+^ T-cell decline, possibly due to higher X4 tropism [40].

Moreover, co-infection and superinfection of the same cells with simian immunodeficiency viruses (SIVs) from different African monkeys, together with recombinant events, may have been relevant for the evolution of SIV infecting chimpanzees (SIV_CPZ_) and for the subsequent adaptation to humans, resulting in the pandemic HIV-1 group M [148].

It is worth mentioning that another potential HIV-1 co-infecting/superinfecting unrelated virus at the cellular level is represented by SARS-CoV-2, the source of the COVID-19 pandemic. Indeed, recent findings provided some evidence for the possible capacity of SARS-CoV-2 to infect blood-circulating leukocytes in COVID-19 patients, while SARS-CoV-2-positive monocytes, B cells, and CD4^+^ T lymphocytes were also detected in postmortem lung tissue [149,150]. Moreover, productive in vitro infection of activated CD4^+^ T cells, but not CD8^+^ T cells, was obtained in an angiotensin-converting enzyme (ACE2)-independent manner, suggesting lymphocyte function-associated antigen 1 (LFA-1) or CD4 as alternative receptors [150,151]. Both in vivo and in vitro data indicate that SARS-CoV-2 infection of CD4^+^ T cells results in impaired CD4^+^ T cell function and cell death [149,150,151]. Collectively, these data provide a possible mechanistic explanation for the lymphocytopenia observed in severe COVID-19 patients and also suggest a common ground, at the cellular level, for SARS-CoV-2 interaction with HIV-1. In this respect, conflicting reports exist about COVID-19 severity and mortality in people living with HIV-1 (PLWH), ranging from increased, especially in patients with lower CD4^+^ T cell counts with or without uncontrolled HIV-1 replication, to reduced, due to a possible anti-SARS-CoV-2 activity of antiretroviral drugs [152,153,154,155,156]. Therefore, future in vivo and in vitro data will be needed to ascertain the outcome and the molecular mechanisms of a possible HIV-1/SARS-CoV-2 interaction within the same cell.

## 6. Conclusions

Nowadays, the possibility to detect co-infections or superinfections between HIV-1 and different viruses may take advantage not only of classic methodologies but also of new investigational tools like single-cell analyses [157,158] coupled with next-generation sequencing [159]. Such technologies provide a unique opportunity to investigate one cell at a time and the phenotype that each cell may represent. Single-cell multi-omics technologies [160] may also provide advances towards a better picture of HIV-1 co-infection/superinfection with other viruses happening in the same cell and the outcome of such close interaction in terms of pathogenesis and treatment management of HIV-1 and the companion virus.

## Figures and Tables

**Figure 1 pathogens-13-00349-f001:**
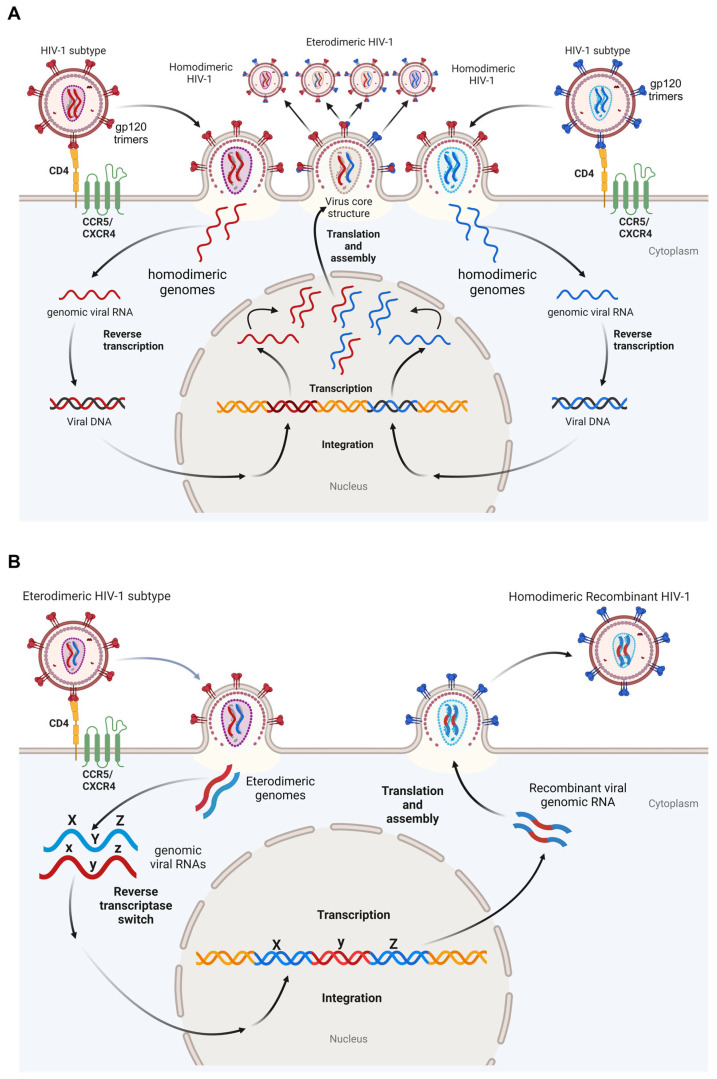
Model for the generation of recombinant forms of HIV-1. (**A**) Following co-infection of the same cell with two different replication-competent HIV-1 subtypes, for example, two copies of genomic RNAs, released from each homodimeric infecting virus, are reverse transcribed into proviral DNA integrating into the host genome. Transcription from the integrated HIV-1 long terminal repeats, leads to the formation of virions harboring both heterodimeric and homodimeric genomes. (**B**) A new infection event involving a heterodimeric HIV-1 virion represents the source of viruses with recombinant genomes, generated by a reverse transcriptase switch event occurring with heterodimeric genomic viral RNAs. Red and blue genomic viral RNAs, and gp120 trimers belonging to different HIV-1 subtypes, are depicted. Heterodimeric and recombinant red and blue genomic viral RNAs and gp120 monomers are also shown. Red/black, and blue/black proviral integrated DNAs are shown. Yellow/orange host genomic double stranded DNA is shown. Upper and lower case x, y, and z letters represent alleles from HIV-1 genomes from different subtypes. Created with Biorender.com.

**Figure 2 pathogens-13-00349-f002:**
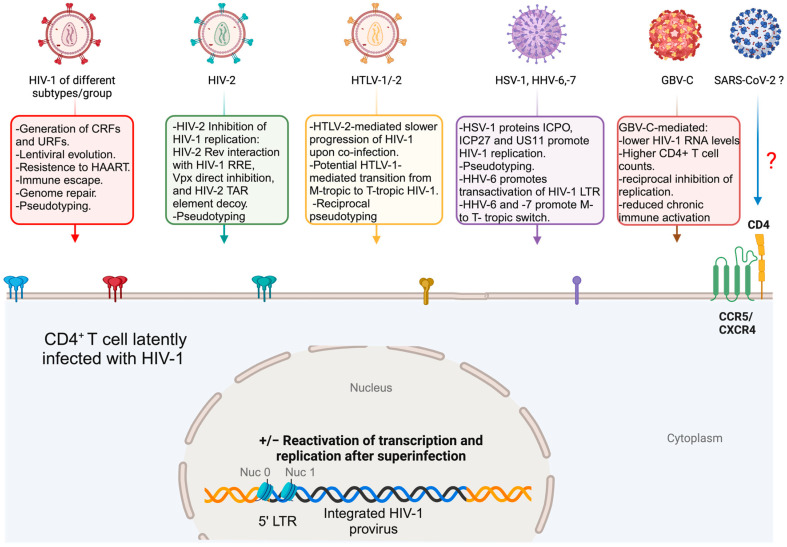
Different superinfecting viruses have the capacity to infect and replicate within the same HIV-1 target cell type, CD4^+^ T cells. The superinfection of HIV-1 latently infected cells with different viruses (i.e., HIV-1 itself, HIV-2, HTLV-1/2, HHVs, and GBV-C) can result in different outcomes, ranging from reciprocal inhibition to increased HIV-1 pathogenesis, and also increased HIV-1 replication. It is not yet clear yet if SARS-CoV-2 can be included in the group of viruses responsible for the co-infection/superinfection with HIV-1 (?). Possible outcomes are highlighted. Colored envelope proteins from resident and superinfecting viruses, involved in pseudotyping, are shown. Blue/black integrated proviral double stranded DNA is shown. Yellow/orange host genomic double stranded DNA is shown. The HIV-1 LTR promoter is indicated between nucleosomes Nuc 0 and Nuc 1. Created with Biorender.com.

**Figure 3 pathogens-13-00349-f003:**
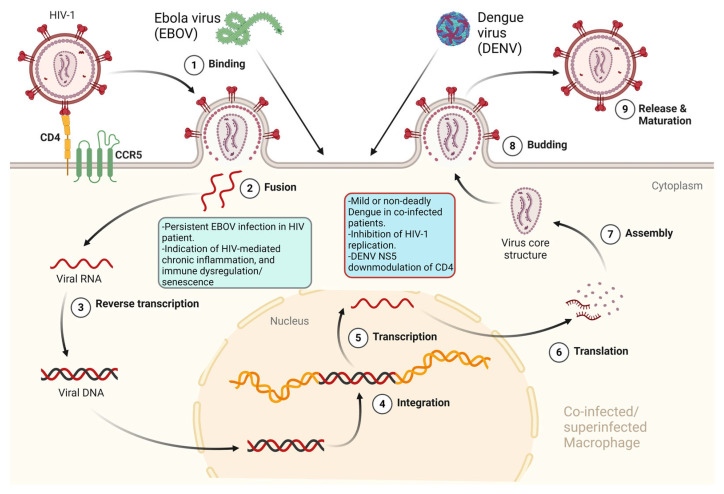
It is feasible to assume that it is highly probable to have co-infections/superinfections between HIV-1 and a widespread virus like Dengue virus (DENV), endemic in tropical and sub-tropical areas of the world, as well as Ebola virus responsible for severe outbreaks in West Africa, in the same individual and in the same cell types (i.e., macrophages) targeted by HIV-1, Dengue, and Ebola viruses. Possible outcomes are highlighted. Red genomic viral RNAs are shown. Red/black proviral integrated double stranded DNA is shown. Yellow/orange host genomic double stranded DNA is shown. Created with Biorender.com.

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
