# Peer review of "Co-Infections and Superinfections between HIV-1 and Other Human Viruses at the Cellular Level"

_pathogens, 2024, doi:10.3390/pathogens13050349_

Round 1

Reviewer 1 Report

Comments and Suggestions for Authors

This submission aims to update readers on the progress in understanding co-infections and superinfections involving HIV and other pathogenic viruses at the cellular level. The manuscript is generally well-written and informative. However, I have an impression that a further improvement is necessary to enhance the quality of the manuscript, as is often the case with comprehensive reviews.

A key suggestion for improvement would be to incorporate schematic diagrams or tables, which could enhance reader comprehension and facilitate a deeper understanding of the review.

In addition, to strengthen the physiological importance of the discussed wide-ranging occurrence of co-infections and superinfections, the authors might contemplate incorporating a section that offers a hypothesis or elucidates the underlying mechanisms.

Author Response

Reviewer 1

Comments and Suggestions for Authors

This submission aims to update readers on the progress in understanding co-infections and superinfections involving HIV and other pathogenic viruses at the cellular level. The manuscript is generally well-written and informative. However, I have an impression that a further improvement is necessary to enhance the quality of the manuscript, as is often the case with comprehensive reviews.

A key suggestion for improvement would be to incorporate schematic diagrams or tables, which could enhance reader comprehension and facilitate a deeper understanding of the review.

In addition, to strengthen the physiological importance of the discussed wide-ranging occurrence of co-infections and superinfections, the authors might contemplate incorporating a section that offers a hypothesis or elucidates the underlying mechanisms.

Response:

we thank the reviewer for the overall appreciation of our review. In response to reviewer’s suggestions, and also to similar suggestions from the other reviewers, we have incorporated three figures. The first, underlines the mechanisms suggested for the formation of recombinant HIV-1 arising from co-infection between different HIV-1 viruses. The second, describes the viruses for which consolidated data exists for a superinfection with an integrated, and possibly latent, HIV-1 in CD4+ T cells. The third one deals with the possible scenario of co-infection between highly pathogenic human viruses in a shared target cell. New sections were included regarding HSV-6/-7 and GBV-C virus, extending the focus also on viruses of low pathogenesis or non-pathogenic.

Reviewer 2 Report

Comments and Suggestions for Authors

This is a very interesting review article that discusses co-infection or superinfection of a cell by two different viruses. The authors focus on HIV and discuss human pathogenic viruses targeting CD4+ T-cells, namely different HIV-1 strains, HIV-2, human T-cell lymphotropic virus, and herpes simplex virus, as well as viruses targeting macrophages and dendritic cells, namely dengue and Ebola viruses. What makes the article interesting is its focus on possible outcomes of such co- and superinfections, such as enhanced replication, inhibited replication, and pseudotyping. As such, I believe the article makes a contribution to the field. The statements and conclusions drawn are coherent and supported by the listed citations.

General comments:

I am not aware of a similar review that has been published recently.

Some of the cited references are very old. While some of them are seminal papers from the 1990s, some can be replaced with more recent publications. I have indicated where the authors might consider other publications of relevance. There is no evidence an excessive number of self-citations.

It has recently been shown that SARS-CoV-2 can also infect CD4+ T-cells by engaging the CD4 receptor – see Brunetti, NS, Davanzo GG, de Moreas D, et al. 2023. SARS-CoV-2 uses CD4 to infect T helper lymphocytes. eLife 12:e84790. There is currently a lot of controversy about this topic and I suggest that the authors include this virus in their discussion, since it is certainly very topical.

The manuscript will also be enhanced by the inclusion of a figure, demonstrating the mechanisms of some of the interactions e.g. inhibition and enhancement.

The text is generally very long  and it would be more readable if the text is broken down into shorter paragraphs.

It would be useful to differentiate between in vitro and in vivo data throughout.

A paragraph about the implications for management of co- or superinfected patients would add value. For instance, some HIV-1 CRFs are known to have different resistance profiles that could impact management. While the authors do state in lines 165-8: “indicating a potentially more relevant impact of co-infection/super-infection-mediated recombination in HIV-1 genetic variability, immune escape, pathogenicity, and antiretrovial therapy failure due to pan-resistant viruses that can be generated starting from multiple drug resistant (MDR) strains” this is part of a long paragraph and may therefore not get the necessary attention. A better strategy might be to expand on this with specific examples and add this as a separate section.   

Specific comments:

The introductory section about HIV (lines 52 – 85) can be shortened since most of it is well known. It would be better to focus only on what is relevant in the context of co- and superinfection. In particular, a brief explanation of why HIV is particularly prone to co- or superinfection would be helpful. While the authors do state in lines 135-7 that “Indeed, the diploid nature of HIV-1 virions as well as reverse transcriptase (RT) switching between the two different RNA strands, both contribute to generate genetic hybrids that may also be relevant for HIV-1 genome repair”, a more detailed explanation would be helpful to sketch the context more clearly.

In addition, it might also add value to refer to the first compelling evidence of lentivirus (SIV) recombination in nature: “Jin  MJ, Hui  H, Robertson  DL, Muller  MC, Barre Sinoussi  F, Hirsch  VM, Mosaic genome structure of simian immunodeficiency virus from West African green monkeys. EMBO J. 1994;13:2935–47.

The section spanning lines 85 to 101 is superfluous, since it is covered again in section 2.

Section 2.1 With regards to co-infection/superinfection between different HIV-1 strains, it would be useful to give a common example, such as the development of CRF_AD or AE forms that have had significant implications in terms of viral spread, disease progression and treatment. See for instance: Chu, M., Zhang, W., Zhang, X. et al. HIV-1 CRF01_AE strain is associated with faster HIV/AIDS progression in Jiangsu Province, China. Sci Rep 7, 1570 (2017). https://doi.org/10.1038/s41598-017-01858-2 and Singh K, Flores JA, Kirby KA, Neogi U, Sonnerborg A, Hachiya A, Das K, Arnold E, McArthur C, Parniak M, et al. Drug Resistance in Non-B Subtype HIV-1: Impact of HIV-1 Reverse Transcriptase Inhibitors. Viruses. 2014; 6(9):3535-3562. https://doi.org/10.3390/v6093535 .

Line 146 – 151. The following sentence is difficult to understand: “In this respect, it has been demonstrated, using latently infected Jurkat T cell lines as well as primary CD4+ T cells, that HIV-1 superinfection results in reactivation of latent HIV-1, depending on the expression of Tat protein from the superinfecting virus, being the H13L Tat variant from the latent virus unable to sustain viral transcription, when expressed beneath a concentration theresold [36].”

Therefore, either HIV-1 viral parti-87 cles, belonging to different subtypes, or HIV-1 and HIV-2 viruses may be present in the 88 same cell together. Ref 46 – 64. In humans?

Line 288. “HSV favours HIV-1 infection and causes severe clinical manifestations in immune compromised hosts, particularly in AIDS patients [80], also favouring HIV-1 infection [81].” These references are very old and should be updated.

Line 329. “similarly to what was reported with measles virus/HIV-1 co-infection [94].”  Why is measles not discussed as an entity as well? The same is true of yellow fever (lines 336-7): “while yellow fever virus (YFV) infection of macrophages was shown to inhibit HIV-1 replication in infected cells also correlating with CD4 downmodulation [98].”

Section 3. Co-infections or superinfections of HIV-1 with human pathogenic viruses targeting macrophages, DCs and other APCs.” It is not clear which “other APCs” are referred to since it seems that only macrophages and dendritic cells are discussed.

Comments on the Quality of English Language

While the text is readable and the meaning is clear, there are a number of grammatical errors and typos that should be corrected to improve the reader experience.  

Author Response

Reviewer 2

This is a very interesting review article that discusses co-infection or superinfection of a cell by two different viruses. The authors focus on HIV and discuss human pathogenic viruses targeting CD4+ T-cells, namely different HIV-1 strains, HIV-2, human T-cell lymphotropic virus, and herpes simplex virus, as well as viruses targeting macrophages and dendritic cells, namely dengue and Ebola viruses. What makes the article interesting is its focus on possible outcomes of such co- and superinfections, such as enhanced replication, inhibited replication, and pseudotyping. As such, I believe the article makes a contribution to the field. The statements and conclusions drawn are coherent and supported by the listed citations.

Response:

we thank the reviewer for the in depth revision and the positive consideration of our review.

Comments and Suggestions for Authors

General comments:

I am not aware of a similar review that has been published recently.

Some of the cited references are very old. While some of them are seminal papers from the 1990s, some can be replaced with more recent publications. I have indicated where the authors might consider other publications of relevance. There is no evidence an excessive number of self-citations.

It has recently been shown that SARS-CoV-2 can also infect CD4+ T-cells by engaging the CD4 receptor – see Brunetti, NS, Davanzo GG, de Moreas D, et al. 2023. SARS-CoV-2 uses CD4 to infect T helper lymphocytes. eLife 12:e84790. There is currently a lot of controversy about this topic and I suggest that the authors include this virus in their discussion, since it is certainly very topical.

Response:

this is a very important suggestion and even if there are no indications to date, to the best of our knowledge, of direct co-infection between HIV-1 and SARS-CoV-2 in the same CD4+ T cell, we discussed the topic in the discussion section.

The manuscript will also be enhanced by the inclusion of a figure, demonstrating the mechanisms of some of the interactions e.g. inhibition and enhancement.

Response:

there are 3 figures now, highlighting HIV-1 recombinants formation and also summarizing each co-infecting virus interaction with HIV-1

The text is generally very long  and it would be more readable if the text is broken down into shorter paragraphs.

Response:

following reviewer’s suggestion we have divided some of the text in further sub-sections.

It would be useful to differentiate between in vitro and in vivo data throughout.

Response:

we appreciate the reviewer’s suggestion, nevertheless we believe it is more helpful to keep it this way to focus the attention on each specific finding, also because in vivo and in vitro data are often present in the same paper cited.

A paragraph about the implications for management of co- or superinfected patients would add value. For instance, some HIV-1 CRFs are known to have different resistance profiles that could impact management. While the authors do state in lines 165-8: “indicating a potentially more relevant impact of co-infection/super-infection-mediated recombination in HIV-1 genetic variability, immune escape, pathogenicity, and antiretrovial therapy failure due to pan-resistant viruses that can be generated starting from multiple drug resistant (MDR) strains” this is part of a long paragraph and may therefore not get the necessary attention. A better strategy might be to expand on this with specific examples and add this as a separate section.   

Response:

we appreciate the reviewer’s suggestion and we discussed implications for management of co- or superinfected patients in the discussion section. We believe that, by dedicating a separate section to this topic, it could go beyond the scope of our review.

Specific comments:

The introductory section about HIV (lines 52 – 85) can be shortened since most of it is well known. It would be better to focus only on what is relevant in the context of co- and superinfection. In particular, a brief explanation of why HIV is particularly prone to co- or superinfection would be helpful. While the authors do state in lines 135-7 that “Indeed, the diploid nature of HIV-1 virions as well as reverse transcriptase (RT) switching between the two different RNA strands, both contribute to generate genetic hybrids that may also be relevant for HIV-1 genome repair”, a more detailed explanation would be helpful to sketch the context more clearly.

Response:

we appreciate the reviewer’s suggestion regarding this specific point and we shortened a little bit the introduction section. Moreover, we dedicated Figure 1 to explain in more details the mechanism behind the generation of HIV-1 recombinant forms upon co-infection.

In addition, it might also add value to refer to the first compelling evidence of lentivirus (SIV) recombination in nature: “Jin  MJ, Hui  H, Robertson  DL, Muller  MC, Barre Sinoussi  F, Hirsch  VM, Mosaic genome structure of simian immunodeficiency virus from West African green monkeys. EMBO J. 1994;13:2935–47.

Response:

we appreciate the reviewer’s suggestion regarding this specific point and evidence of lentivirus (SIV) recombination in nature are now present throughout the text together with the suggested paper.

The section spanning lines 85 to 101 is superfluous, since it is covered again in section 2.

Response:

we appreciate the reviewer’s suggestion, nevertheless we believe that an brief introduction of all viruses reported in the review would help the reader for the subsequent detailed descriptions.

Section 2.1 With regards to co-infection/superinfection between different HIV-1 strains, it would be useful to give a common example, such as the development of CRF_AD or AE forms that have had significant implications in terms of viral spread, disease progression and treatment. See for instance: Chu, M., Zhang, W., Zhang, X. et al. HIV-1 CRF01_AE strain is associated with faster HIV/AIDS progression in Jiangsu Province, China. Sci Rep 7, 1570 (2017). https://doi.org/10.1038/s41598-017-01858-2 and Singh K, Flores JA, Kirby KA, Neogi U, Sonnerborg A, Hachiya A, Das K, Arnold E, McArthur C, Parniak M, et al. Drug Resistance in Non-B Subtype HIV-1: Impact of HIV-1 Reverse Transcriptase Inhibitors. Viruses. 2014; 6(9):3535-3562. https://doi.org/10.3390/v6093535 .

Response:

we appreciate the reviewer’s suggestion regarding this specific point together with the suggested papers and CRF AE example, together with the implications for disease progression and treatment is now present in the discussion section.

Line 146 – 151. The following sentence is difficult to understand: “In this respect, it has been demonstrated, using latently infected Jurkat T cell lines as well as primary CD4+ T cells, that HIV-1 superinfection results in reactivation of latent HIV-1, depending on the expression of Tat protein from the superinfecting virus, being the H13L Tat variant from the latent virus unable to sustain viral transcription, when expressed beneath a concentration theresold [36].”

Response:

we appreciate the reviewer’s suggestion regarding this specific point and we agree with the reviewer’s concern about the clarity of the sentence, therefore it was simplified.

Therefore, either HIV-1 viral particles, belonging to different subtypes, or HIV-1 and HIV-2 viruses may be present in the same cell together. Ref 46 – 64. In humans?

Response:

we did not fully understand this specific reviewer comment but we tried to add clarifications regarding this sentence anyway.

Line 288. “HSV favours HIV-1 infection and causes severe clinical manifestations in immune compromised hosts, particularly in AIDS patients [80], also favouring HIV-1 infection [81].” These references are very old and should be updated.

Response:

we appreciate the reviewer’s suggestion regarding this specific point and more recent references were added to support the sentence highlighted by the reviewer.

Line 329. “similarly to what was reported with measles virus/HIV-1 co-infection [94].”  Why is measles not discussed as an entity as well? The same is true of yellow fever (lines 336-7): “while yellow fever virus (YFV) infection of macrophages was shown to inhibit HIV-1 replication in infected cells also correlating with CD4 downmodulation [98].”

Response:

we appreciate the reviewer’s suggestion regarding this specific point. We did not want to discuss Measles and YFV because of the fact that these pathogens have been preventable by vaccination for a long time now differently from Dengue virus that, only very recently has vaccines available. We clarified this in the text. On the other side, we added new sections dealing with HSV-6 and 7 and also the flavivirus GBV-C, because we believe that there are solid evidence for co-infection/superinfection with HIV-1 within the same CD4+ T cells. By doing this we expanded the focus of the review, focusing not only to highly pathogenic co-infecting viruses but also to low pathogenic and non-pathogenic viruses.

Section 3. Co-infections or superinfections of HIV-1 with human pathogenic viruses targeting macrophages, DCs and other APCs.” It is not clear which “other APCs” are referred to since it seems that only macrophages and dendritic cells are discussed.

 Response:

we appreciate the reviewer’s suggestion and the sentence was modified by eliminating “APCs”.

Comments on the Quality of English Language

While the text is readable and the meaning is clear, there are a number of grammatical errors and typos that should be corrected to improve the reader experience.  

Response:

we apologize for this inconvenience that was mainly due to a problem with the specific Microsoft Word original file, that for some reason was not able to detect English typos originated during a “fast writing”.

Reviewer 3 Report

Comments and Suggestions for Authors

The review by Acchioni and colleagues entitled “Co-infections and superinfections between HIV and other pathogenic viruses at the cellular level” is an interesting collection of facts known about mutual effects of HIV and other viruses infecting the same cell. It is well written and mostly easy to understand.

These are my criticisms:

1.       Line 62: The sentence “Through the binding of the viral Env glycoprotein gp120 to the CD4 receptor, and interaction of viral gp41 to either CCR5 or CXCR4 cellular co-receptors, HIV-1 is able to productively infect CD4+ lymphocytes …..” needs clarification. Do the authors mean that gp41 interacts with CCR5 and CXCR4? To my knowledge current models imply an interaction of gp120 and not gp41 with the co-receptors. If an interaction of gp41 with the coreceptors is published, the reference should be cited.

2.       Line 88: HIV-1 belonging to different groups can also be present in a single cell. The sentence reporting that HIV of different subtypes and HIV-2 can be in a single cell should also mention HIV of two different groups as reason for co-infection/superinfection.

3.       Line 220: Please clarify what “90% inhibition at 0.5 molar ratio” means.

4.       I wonder why the superinfection of HIV and KSHV (HHV-8) leading to Karposi Sarcomas is not mentioned at all in the review. It is clinically very relevant although the tropism if the viruses does presumably not overlap. I think it should at least be shortly mentioned.

5.       The review would benefit from one or two figures. In one figure it could be shown, how CRF/URFs of HIV are generated. A second figure could show the steps of the HIV replication cycle where the stated coinfecting/superinfecting viruses interfere. This is only a suggestion.

Minor critics:

There are several typos in the text: e.g. line 187 (asimptomatic should be asymptomatic), line 196 (isolated in should be changed to isolated from), line 203 (replicated into should changed to replicate in), line 263 (solble should be soluble), line 264 (sinergy should be synergy).

Comments on the Quality of English Language

Editing of English needed.

Author Response

Reviewer 3

The review by Acchioni and colleagues entitled “Co-infections and superinfections between HIV and other pathogenic viruses at the cellular level” is an interesting collection of facts known about mutual effects of HIV and other viruses infecting the same cell. It is well written and mostly easy to understand.

We thank the reviewer for the appreciation of our review

Comments and Suggestions for Authors

These are my criticisms:

  1. Line 62: The sentence “Through the binding of the viral Env glycoprotein gp120 to the CD4 receptor, and interaction of viral gp41 to either CCR5 or CXCR4 cellular co-receptors, HIV-1 is able to productively infect CD4+ lymphocytes …..” needs clarification. Do the authors mean that gp41 interacts with CCR5 and CXCR4? To my knowledge current models imply an interaction of gp120 and not gp41 with the co-receptors. If an interaction of gp41 with the coreceptors is published, the reference should be cited.

Response:

we appreciate the reviewer’s suggestion regarding this specific point. As stated in the original submitted file at line 175-178 “The binding of the HIV-1 Envelope glycoprotein gp120, protruding from the envelope of HIV-1 viral particles, with CD4 produces conformational modifications of gp120, leading to its interaction with CCR5 or CXCR4 cellular coreceptor, with the subsequent membrane fusion, mediated by HIV-1 glycoprotein gp41, and HIV-1 entry into target cells”, we are fully aware of the fact that is gp120 that dictate co-receptor preference. The sentence the reviewer mentioned originated by a mistaken assembly of the sentence it self that went unnoticed by us and was therefore corrected. Anyway, we truly appreciate reviewer’s scrutiny of our work that avoided such mistake to go unchecked.

  1. Line 88: HIV-1 belonging to different groups can also be present in a single cell. The sentence reporting that HIV of different subtypes and HIV-2 can be in a single cell should also mention HIV of two different groups as reason for co-infection/superinfection.

Response:

we appreciate the reviewer’s suggestion regarding this specific point. The sentence was corrected accordingly.

  1. Line 220: Please clarify what “90% inhibition at 0.5 molar ratio” means.
  2.  

Response:

we appreciate the reviewer’s suggestion regarding this specific point. The original sentence in the reported paper refers to the molar ratio of 0.5 ( 1 microg of HIV-2 provirus DNA transfected  together with 2 microg of HIV-1 provirus DNA, assuming the same size in bp between the two, therefore they used the term “molar ratio”) compared to the “relative HIV-1 expression” measuring p24 protein and viral RNA.

  1. I wonder why the superinfection of HIV and KSHV (HHV-8) leading to Karposi Sarcomas is not mentioned at all in the review. It is clinically very relevant although the tropism if the viruses does presumably not overlap. I think it should at least be shortly mentioned.

Response:

we appreciate the reviewer’s suggestion regarding this specific point and in response to this we discussed HHV-8 HIV-1 interactions in the new introduction sub section “1.3 Potential HIV-1 co-infecting/superinfecting viruses at the cellular level”.

  1. The review would benefit from one or two figures. In one figure it could be shown, how CRF/URFs of HIV are generated. A second figure could show the steps of the HIV replication cycle where the stated coinfecting/superinfecting viruses interfere. This is only a suggestion.

Response:

we appreciate the reviewer’s suggestion regarding this specific point and, as suggested, we added 3 figures following reviewer’s indications.

Minor critics:

There are several typos in the text: e.g. line 187 (asimptomatic should be asymptomatic), line 196 (isolated in should be changed to isolated from), line 203 (replicated into should changed to replicate in), line 263 (solble should be soluble), line 264 (sinergy should be synergy).

Comments on the Quality of English Language

Editing of English needed.

Response:

we apologize for this inconvenience that was mainly due to a problem with the specific Microsoft Word original file, that for some reason was not able to detect English typos originated during a “fast writing”.

Round 2

Reviewer 1 Report

Comments and Suggestions for Authors

The authors have addressed my remarks.